# Phenolic Acid Content and Antioxidant Properties of Edible Potato (*Solanum tuberosum* L.) with Various Tuber Flesh Colours

**DOI:** 10.3390/foods12010100

**Published:** 2022-12-25

**Authors:** Tomasz Cebulak, Barbara Krochmal-Marczak, Małgorzata Stryjecka, Barbara Krzysztofik, Barbara Sawicka, Honorata Danilčenko, Elvyra Jarienè

**Affiliations:** 1Department of Food Technology and Human Nutrition, Institute of Food Technology and Nutrition, College of Natural Sciences, University of Rzeszów, 35-601 Rzeszów, Poland; 2Department of Plant Production and Food Safety, Carpathian State College in Krosno, 38-400 Krosno, Poland; 3The Institute of Human Nutrition Science and Agricultural, University College of Applied Sciences in Chełm, 22-100 Chełm, Poland; 4Department of Plant Production Technology and Commodity Sciences, University of Life Sciences, 20-950 Lublin, Poland; 5Department of Plant Biology and Food Sciences, Vytautas Magnus University, Agriculture Academy, LT-53361 Kauno, Lithuania

**Keywords:** coloured potatoes, phenolic acids, polyphenols, antioxidant properties, anthocyanins

## Abstract

The aim of the paper was to evaluate the phenolic acid content and antioxidant properties in potato cultivars with coloured flesh and bright flesh. The study material included eight cultivars of edible potato grown in a temperate climate in Poland. Five cultivars were potato tubers with coloured flesh: “Rote Emma”, “Blue Salad”, “Vitelotte”, “Red Emmalie”, and “Blue Congo”; and three were potato tubers with bright flesh: “Bella Rosa”, “Lord”, and “Tajfun”. In all potato samples under study, four phenolic acids were identified: chlorogenic acid, gallic acid, p-coumaric acid, and caffeic acid. The predominant acid was chlorogenic acid, the levels of which ranged from 62.95 mg·100 g^−1^ FM to 126.77 mg·100 g^−1^ FM. The total concentration of the identified phenolic acids was diverse and depended on the genotype of the cultivar and the tuber flesh colour, with coloured-fleshed potatoes having higher phenolic acid contents in comparison to bright-fleshed potato cultivars. The average concentration of phenolic acids in the samples was 89.19 mg∙100 g^−1^ FM, and the average concentrations of the individual phenolic acids identified were as follows: chlorogenic acid (86.19 mg∙100 g^−1^ FM), gallic acid (1.18 mg∙100 g^−1^ FM), p-coumaric acid (0.64 mg∙100 g^−1^ FM), and caffeic acid (1.18 mg∙100 g^−1^ FM). In addition, three groups of anthocyanins were identified: pelargonidin-3,5-di-O-glucoside, peonidin-3,5-di-O-glucoside, and petunidin-3,5-di-O-glucoside. Anthocyanins were not found in the “Lord” or “Tajfun” varieties characterised by white tuber flesh. The predominant pigment was petunidin-3,5-di-O-glucoside, with an average content of 23.15 mg∙100 g^−1^ FM, and the highest value was observed in the “Vitelotte” variety (51.27 mg∙100 g^−1^ FM). The antioxidant activity of the flesh of the potatoes under study was diverse depending on flesh colour. The FRAP (Ferric Reducing Antioxidant Power) assay indicated higher antioxidant activity of coloured-fleshed potato cultivars. The highest concentration was identified in the “Vitelotte” cultivar flesh and was 114% higher than in the “Lord” cultivar. Similar dependencies were found in the case of the DPPH (2,2-diphenyl-1-picrylhydrazyl) assay; however, in this case, the “Vitelotte” cultivar flesh demonstrated nearly 6.4 times higher antioxidant activity than the “Lord” cultivar. Summarizing our own research, we can conclude that potato varieties with coloured flesh are characterised by a higher content of biologically active substances, including phenolic acids, and antioxidant properties compared to potato tubers with bright flesh.

## 1. Introduction

Potato (*Solanum tuberosum* L.) is one of the most popular foods consumed around the globe, following rice and wheat [1]. Its nutritional value is based on the high content of carbohydrates—mainly starch—and high-quality protein, macro- and micro-elements, and bioactive ingredients [2,3]. Scientists studying potato quality have paid particular attention to coloured-skinned and coloured-fleshed potatoes, which are still little-known and not very popular raw materials in terms of use in households and food processing [4,5]. In the tested genetic materials of coloured-fleshed and -skinned potatoes, chemical compounds were found that protect human cells from damage caused by free radicals, prevent oxidised low-density lipoprotein cholesterol, and contribute to a lower incidence of some types of cancer, neurodegenerative diseases, osteoporosis, and diabetes [6,7]. The health benefits of coloured-fleshed and -skinned potatoes were corroborated by Akyol et al. [8], Kazimierczak et al. [9], and Liao et al. [10], who identified polyphenols in potato. Most of those polyphenols were phenolic acids—mainly chlorogenic acid and, to a lesser extent, caffeic, cinnamic, p-coumaric, ferulic, and sinapinic acid—as well as flavonoids—mainly catechin and epicatechin. The presence of polyphenols in the human diet is crucial in the prevention of a number of lifestyle diseases [4,11,12,13,14]. According to those authors, polyphenolic compounds in the diet can help maintain good body condition, prevent many diseases, and facilitate the treatment of existing disorders. According to Zhang et al. [14] and Koszowska et al. [11], a diet rich in antioxidants is beneficial to health as it decreases the incidence of cardiovascular diseases, diabetes, cancers, and osteoporosis. Research by Brown et al. [15] and Lachman et al. [6] showed that coloured-fleshed and -skinned potato cultivars also contain flavonol aglycones, which are potent antioxidants. These compounds are responsible for the antioxidant properties of potatoes. As compared to white- or yellow-fleshed cultivars, coloured-fleshed and -skinned potatoes contain almost three times as many polyphenolic compounds, including anthocyanins, which are not present in traditional cultivars. The authors believe that red-fleshed potatoes contain acylated glycosides of pelargonidin, whereas purple-fleshed potatoes contain acylated glycosides of malvidin, petunidin, peonidin, and delphinidin. In their research, Piątkowska et al. [16] reported that products containing anthocyanin compounds are beneficial to health and have anti-atherosclerotic, anti-inflammatory, antioxidant, and anti-cancer properties. According to Zhang et al. [14] and Koszowska et al. [11], anthocyanins have the ability to prevent brittleness of blood vessels and capillaries and can stimulate rhodopsin production, which is necessary in the vision process. In addition, they have a positive influence on decelerating the oxidation of LDL cholesterol, which constitutes atheromatous plaque. The health benefits of anthocyanins were also confirmed by the research by Zawistowski et al. [17], and Zhang et al. [14], who reported that anthocyanins not only have protective properties in case of neoplastic changes but also increase insulin sensitivity and have a beneficial effect on the lipid profile of the system, resulting in its decrease. Research by Jiang et al. [18] showed that anthocyanins present in purple-fleshed potatoes participate in liver regeneration following alcohol-related damage. The multi-directional anthocyanin activity in the system gives hope for its preventive or medicinal use in the treatment of many diseases [4,12,13,14,18]. In addition, a study by Han et al. [19] on rats that were fed purple potato flakes showed that the flakes had antioxidant functions with respect to radical scavenging and inhibition of linoleic acid oxidation and improved antioxidant potential in the rats by increasing hepatic mRNA expression of Mn-SOD, Cu/Zn-SOD, and GHS-Px. Other authors’ studies have shown that coloured-fleshed potatoes demonstrate high antioxidant activity and have the potential to reduce oxidative stress [1,11,15,20,21,22,23,24]. Coloured-fleshed potatoes are recommended for consumption and the production of fried and dried products, particularly due to their good organoleptic characteristics (flavour, smell, texture, and colour) and their higher content of biologically active ingredients [4,20]. The introduction of coloured-fleshed potatoes into food processing could improve potato product diversity, which would stand out with not only their colour but also their higher content of biologically active compounds. Therefore, the consumption of coloured-fleshed potato cultivars can potentially have more health benefits than traditional, white-fleshed cultivars. It can have health benefits related to antioxidants, such as anti-cancer, anti-ageing, and anti-inflammatory properties. Source literature reports that coloured-fleshed potatoes are beneficial to health. However, there has been little research on the influence of potato cultivars on the level of phenolic acids in and antioxidant potential of potatoes with coloured and bright flesh. Therefore, the aim of this paper was to gain such knowledge and raise consumer awareness of the health benefits of coloured potato varieties.

## 2. Materials and Methods

### 2.1. Plant Material

The raw material used in this research involved tubers of 8 edible potato cultivars. Five cultivars were potato tubers with coloured flesh: “Rote Emma”, “Blue Salad”, “Vitelotte”, “Red Emmalie”, and “Blue Congo”; and three were potato tubers with bright flesh: “Bella Rosa”, “Lord”, and “Tajfun”. A description of cultivar characteristics is included in Table 1. The edible potato tubers were obtained from the plantation of the Carpathian State College in Krosno (latitude 21°46′ E; longitude 49°42′ N), Poland. The field experiment was conducted in 2021 in slightly acidic brown earth (pH/KCL 5.67) [25]. The concentration of assimilable phosphorus was high (12.3 mg·100 g^−1^); the concentration of potassium was medium (20.2 mg·100 g^−1^); and the concentration of magnesium was very high (19.5 mg·100 g^−1^). However, the content of copper, manganese, iron, and zinc in the soil was medium (Cu, 5.64 mg·100 g^−1^; Fe, 1.574 mg·100 g^−1^; Mn, 176 mg·100 g^−1^; and Zn, 14.3 mg·100 g^−1^). The average concentration of humus in the topsoil amounted to 2.72%. The results of soil analysis were evaluated based on limit values established by the Institute of Soil Science and Plant Cultivation of the National Research Institute in Puławy [26]. The field study was based on a randomised block design in three replicates. In the autumn, farmyard manure was applied at a rate of 25 t·ha^−1^ in addition to mineral fertilisers used at the following rates: 35 kg·ha^−1^ P (in the form of 46% triple superphosphate), 100 kg·ha^−1^ K (in the form of 60% potassium salt), and 80 kg N per 1 ha (in the form of 34% ammonium saltpeter); nitrogen was applied in the spring [27]. Potatoes were planted at 67.5 cm × 37 cm spacing in mid-April and harvested in September. The size of crop plots was 22.5 m^2^. To control weeds, a mixture of the herbicides Command 480 SC 0.2 dm^3^·ha^−1^ + Afalon Dyspersyjny 450 SC 1.0 dm^3^·ha^−1^ was applied 5–7 days prior to potato plant emergence [27]. Potato blight was controlled using Ridomil Gold MZ 68 WG and Dithane 455 SC (Syngenta, basle, switzerland), and Colorado potato beetle was controlled by means of Apacz 50 WG and Actara 25 WG (Syngenta) [27]. Precipitation and air temperature in that period differed between the potato vegetation months. The year 2021 was warm with excessive precipitation in April, May, and July but significant deficits in July, August, and September.

### 2.2. Sample Preparation

Twenty tubers were collected at random, washed and dried, cut into 10 mm thick slices, and then frozen at −35 °C. The samples were dehydrated with the Sublimator (ZIRBUS Technology GmbH, Bad Grund, Germany), ground in a knife mill (Grindomix GM 200, Retsch GmbH, Haan, Germany), and stored in a refrigerator in sealed plastic bags [4].

### 2.3. Polyphenol Extraction and Identification

For the purpose of identifying phenolic compounds in potato extracts, High-Performance Liquid Chromatography (HPLC) was applied, as previously described [9,27]. The sample preparation procedure involved the extraction of phenolic compounds from dehydrated potato samples (100 mg) with 80% methanol in plastic tubes by means of an ultrasonic bath (10 min, 30 °C). Then, the samples were centrifuged (10 min, 4000× *g*, at 4 °C). Then, 900 µL of supernatant was injected into the HPLC vial, and 100 µL was injected on the HPLC column (Phenomenex, Fusion-80A, C-18, practical shape 4 µm, 4.6 mm × 250 mm, Nexera-i Plus Method Transfer, Shimadzu, Kyoto, Japan). Chromatographic separation was performed at 33 ± 1 °C with the following solvents: a mixture of water and acetonitrile (10% in phase A and 55% in phase B) at the flow rate of 1 mL min^−1^ used as a gradient solvent (1.00–22.99 min phase A 95%; 23.00–27.99 min phase A 50%; 28.00–28.99 min phase A 80%; 29.00–35.99 min phase A 80%; 36.00–38.00 min phase A 95%). The wavelength used for detection ranged from 270–360 nm. External polyphenol reference materials from the Sigma-Aldrich company were used in the analysis, with a purity of 95.00–99.99%. Phenolic compounds were identified based on reference materials and retention times [4,12]. The following wavelengths were used: 250 nm for phenolic acids (gallic, chlorogenic, caffeic, and p-coumaric) and 370 nm for flavonoids (quercetin-3-O-rutinoside, quercetin-3-O-glucoside, and quercetin) [4,12]. The results were expressed in mg 100 g^−1^ FM.

### 2.4. Anthocyanin Extraction and Identification

Whole potatoes were washed in cold water, dried, and cut into small pieces by means of a vegetable chopper. The study material of 800 g was blanched for 3 min in 800 mL of water. The same amount of a mixture of water and hydrochloric acid (19/1, v:v) was cooled to 0 °C for 3 h and stored at 0 °C. Then, it was stored at room temperature for at least 8 h. In order to remove any solid material, the suspension was filtered through glass wool. The raw extract was applied on the Amberlite XAD-7 column. The column was carefully washed with water, and anthocyanins were eluted with a methanol/acetic acid mixture (19/1, v:v). The eluate was concentrated on a vacuum evaporator, then diluted in water and dehydrated. Anthocyanins were measured by means of the HPLC method according to the protocol previously described by Rodriguez-Saona et al. [28], subject to minor modifications. Approximately 50 mg of fine-ground dehydrated potato tissue was added to 50 mL centrifuge tubes. Then, 10M HCl (hydrochloric acid; 2.5 mL) and pure 100% methanol (5 mL) were added, vortexed, and soniced for 30 sec and left in dimmed light in a cool place for 15 min, stirred occasionally. Then, the sample was centrifuged at 6000× *g* for 15 min. Then, 1 mL of the extract was transferred into the HPLC vial (Nexera-i Plus Method Transfer, Shimadzu, Japan), and 100 µL was injected onto the analytical column (Phenomenex, Fusion-80A, C-18, practical shape 4 µm, 4.6 mm × 250 mm). Two solvent systems were used. Solvent system A included water/acetonitrile/formic acid 87/3/10 (v:v:v), whereas solvent system B included water/acetonitrile/formic acid 40/50/10 (v:v:v). The gradient was as follows: 0 min, 6% B; 20 min, 20% B; 35 min, 40% B; 40 min, 60% B; 45 min, 90% B; and 55 min, 6% B. The analysis was carried out with a wavelength of 530 nm, and the analysis duration was 18 min. Anthocyanins (pelargonidin-3,5-di-O-glucoside, peonidin-3,5-di-O-glucoside, and petunidin-3,5-di-O-glucoside) were identified based on their pure reference materials and retention time.

### 2.5. Extract Preparation

The extraction of total phenols and anthocyanins was conducted based on the method by Yang et al. [29], subject to minor modifications. Two grams (g) of dehydrated potato sample was added to 15 mL of extraction solvent (95% alcohol) acidified with 1.5 N HCl (85:15, v/v). The tube was flushed twice. The obtained mixture was topped up to 50 mL and extracted for 12 h at 4 °C. The supernatant was collected into a 50 mL conical tube, and then centrifuged (4000 rpm, 10 min, 4 °C) in a centrifuge (ST16R, Thermo Scientific, Waltham, MA, USA). Polar lipids and other interfering compounds were removed by means of a method by Song et al. [7]. In total, 18 mL of hexane was added to 6 mL of the obtained extract and mixed thoroughly prior to removal of the hexane layer. This step was repeated six times until the hexane layer was removed completely. The chlorophyll-free extract was filtered with a 0.22 µM organic membrane and then used in the analysis of phenolic and anthocyanin content and the antioxidant activity.

### 2.6. Measuring Total Phenolic and Anthocyanin Content

Total phenolic content (TPC) was measured via the Folin–Ciocalteu method with slight modifications. A total of 2 mL of the Folin–Ciocalteu reagent was added to 1 mL of the extract and mixed thoroughly. After 5 min, 2 mL of sodium carbonate was added (10 g/100 mL). The obtained mixture was mixed thoroughly and stored in darkness for 1 h at room temperature, then absorbance was measured with a wavelength of 760 nm by means of the UV/Visible spectrophotometer (DU800, Beckman Coulter, Brea, CA, USA). TPC was calculated using a calibration curve of gallic acid within the range from 1.045 to 10.45 µg/mL, expressed as a milligram of gallic acid equivalent per 100 g of fresh mass (mg GAE/100 g FM). Total anthocyanin content (TAC) was measured by means of a method described by Fuleki and Francis [29]. The extract samples were mixed at a 1:20 (v:v) ratio with buffers: potassium chloride and sodium acetate with the pH of 1.0 and 4.5, respectively, in separate tubes. Fifteen minutes later, absorbance of each solution was measured at a wavelength of 535 nm. Total anthocyanin content was calculated according to the following formula:TAC (mg/100 g) = RG_535_ × V × N ÷ 98.2 ÷ m × 100(1)
where RG_535_, V, N, 98.2, and m refer to the following, respectively: the spectrophotometric reading at a wavelength of 535 nm, extract volume, dilution factor, extinction coefficient, and sample mass.

### 2.7. DPPH Radical-Scavenging Activity

The ability of the samples under study to scavenge the stable free radical 2,2-diphenyl-1-picrylhydrazyl (DPPH, Sigma Aldrich Co., St. Louis, MO, USA) was measured by means of method [30]. An amount of 10 mL of distilled water (HLP 20UV, HYDROLAB, Straszyn, Poland) and 1.8 mL of a solution of 0.1 mM of DPPH in methanol (Sigma Aldrich Co., USA) was added to 2 g of the study sample. The obtained mixture was left in darkness for 1 h at room temperature. Then, the absorbance of the mixture was measured spectrophotometrically (UV-2600i Spectrophotometer, Shimadzu, Japan) at 517 nm, with methanol as blind feed. The calibration curve was delineated within the range from 0.1 to 100 μgmL^−1^ of Trolox solution (Sigma-Aldrich, Co., St. Louis, MO, USA) in ethanol. The calculation was expressed as [μmol Trolox ∗ g^−1^ DW].

### 2.8. ABTS Radical-Scavenging Activity

The ABTS radical-scavenging activity was measured with a method by Re et al. [31] with minor modifications described by Liao et al. [10]. At first, 4 µL of extract and 36 µL of absolute ethanol were poured onto a 96-well microplate with a flat bottom, and then 200 µL of ABTS radical solution was added. After mixing thoroughly and leaving the mixture in darkness for 6 min, absorbance was measured (Multiskan SkyHigh, Thermo Scientific, USA) at a wavelength of 734 nm. The results were expressed as µM of Trolox equivalent per gram of dry weight (µM Trolox ∗ g^−1^ DW).

### 2.9. Ferric Reducing Antioxidant Power (FRAP) Assay

The FRAP assay was conducted according to the method described by Du et al. [32], subject to minor modifications. First, 0.2 mL of PBS and 1.5 mL of 0.3% (w/v) potassium ferrocyanide were added to 1.0 mL of extract diluted 10-times and incubated at 50 °C for 20 min. Then, 1.0 mL 10% (w/v) of trichloroacetic acid was added. Next, the obtained samples were centrifuged for 10 min at 4000 rpm at a temperature of 4 °C. Afterwards, 2.0 mL of the supernatant was collected, and 0.5 mL of 0.3% (w/v) iron trichloride and 3.0 mL of distilled water were added. Absorbance was measured with the spectrophotometer (UV-2600i, Shimadzu, Japan) at 700 nm. The results are expressed as μmol Trolox g^−1^ DW.

### 2.10. Measuring Total and Reduced Ascorbate Content

Ascorbate (as ascorbic acid, AsA) was extracted and analysed in fresh potato tubers based on the method developed by Bartoli et al. [33], subject to minor modifications. Approx. 5 g of the potato sample under study was mixed with 5 mL of 5% metaphosphoric acid, and then ground in a pre-cooled ball mill (Teflon bowl with two zirconium balls) for 2 min. The homogenates were decanted into a 50 mL plastic centrifuge tube, and then another 10 mL of 5% metaphosphoric acid was used to flush the Teflon bowl and zirconium balls, and decanting was carried out into the same 50 mL plastic centrifuge tube. The homogenates were centrifuged at 10,000 rpm for 15 min at 4 °C. The supernatant was poured into a cooled clean 50 mL plastic centrifuge tube and placed in ice. In order to measure the reduced ascorbate content, 1000 µL of 150 mM phosphate buffer (pH 7.4; 5 mM EDTA) and 200 µL of Milli-Q water (Millipore, Bedford, MA, USA) were added to 200 µL of the supernatant [33]. In the case of the total ascorbate content, the same procedure was applied, with the only difference being that 200 µL of Milli-Q water was replaced with 200 µL of 5 mM dithiothreitol. The reference material preparation was the same as in the case of measuring reduced ascorbate content, with the only difference being that 200 µL of each reference material was replaced with the relevant sample amount. The samples and reference materials were incubated at room temperature for 15 min in darkness. To each sample and reference material, o-phosphoric acid (100 µL) was added in order to neutralise dithiothreitol and acidify the solution for the analysis based on High-Performance Liquid Chromatography (HPLC) [33]. Then, the samples were filtered through 0.2 µm polyvinylidene fluoride filters (Chromatographic Specialties Inc., Brockville, Canada) mounted on a glass syringe, which was flushed three times with methanol in-between the samples. Reduced and total ascorbate contents were measured quantitatively by means of HPLC (Nexera-i Plus Method Transfer, Shimadzu, Japan) with the C18 analytical column (Luna 5 µL 150 × 4.6 mm i.d., Phenomenex, Torrance, CA, USA) [33]. Oxidised AsA is the difference between total ascorbate content and reduced ascorbate content.
Oxidised AsA = Total AsA − Reduced AsA(2)

### 2.11. Statistical Analysis

The results are presented as the mean of three independent measurements. All analyses were performed in Statistica v. 13.3 (StatSoft, Krakow, Poland). The significance of the means was examined post-hoc in Tukey’s test, which is part of the one-way analysis of variance (ANOVA). Differences of *p* < 0.05 were considered significant. In order to describe the mutual relations between the variables under study, variable reduction techniques based on scaled heat maps made in the procedures of the R studio program were used.

## 3. Results and Discussion

### 3.1. Phenolic Acid Content

The study showed that the phenolic acid content in potato varieties with coloured flesh was higher than in varieties with bright flesh (Table 2). The highest phenolic acid contents among the coloured-fleshed tubers were reported in the “Vitelotte” and “Blue Salad” cultivars (129.94 and 120.62 mg·100 g^−1^ FM, respectively). Among the bright-fleshed cultivars, higher phenolic acid contents were determined in the “Bella Rosa” and “Tajfun” cultivar tubers (66.50 and 65.30 mg 100 g^−1^ FM, respectively). Our own research showed that the genetic properties of the cultivars under study and the flesh colour had a significant influence on phenolic acid content in bright- and coloured-fleshed potato tubers (Table 2).

Vaitkevičienė et al. [4] arrived at similar results in their study. In their and our own research, phenolic acid content in potato tubers depended on genetic variation (cultivar). Research by other authors has shown that the genetic content of potato cultivars had more influence on phenolic content than the environment [2]. According to De Masi [2], increased total phenolic acid content in coloured potatoes is related to anthocyanin content, due to which the tubers are strongly pink, purple, or blue.

According to Hosseini-Beheshti et al. [34], differences in the antioxidant properties of coloured cultivars probably result from not only total phenolic acid content but also differences in anthocyanin stabilisation by other compounds found in tuber dry mass and different antioxidant potentials of various anthocyanins.

In the research, four phenolic acids (chlorogenic acid, gallic acid, p-coumaric acid, and caffeic acid) were identified in edible potato tubers. Their concentration in descending order was as follows: chlorogenic acid > gallic acid > caffeic acid > p-coumaric acid (Table 2). The tests conducted on those acids showed significant differences in their content in tubers depending on flesh colour, with chlorogenic acid being the dominating one in all cultivars, ranging from 62.95 mg·100 g^−1^ to 126.77 mg·100 g^−1^ FM in the samples under study. The highest levels of chlorogenic acid were found in coloured-fleshed tubers, and the lowest in bright-fleshed potatoes (Table 2). The results of chlorogenic acid content were also confirmed in the research by Vaitkevičienė et al. [4], who found that chlorogenic acid was dominant in all potato cultivars under study. Franková et al. [13] also confirmed in their research that purple-fleshed potato tubers have a high content of chlorogenic acid. In our own research, the content of gallic acid in the samples under study was determined at a level from 0.69 mg·100 g^−1^ FM in the “Lord” cultivar (white-fleshed) to 1.55 mg·100 g^−1^ FM in the “Rote Emma” cultivar (coloured-fleshed). The highest p-coumaric acid content was found in the “Rote Emma” cultivar (0.79 mg·100 g^−1^ FM, coloured-fleshed), whereas the lowest was found in the bright-fleshed “Bella Rosa” cultivar (0.48 mg·100 g^−1^ FM on average). The caffeic acid content ranged from 0.58 mg·100 g^−1^ FM (the bright-fleshed “Lord” cultivar) to 2.55 mg·100 g^−1^ FM (the coloured-fleshed “Rote Emma” cultivar) (Table 2). According to other authors [35,36], chlorogenic acid is dominant among potato phenolic acids, and its concentration is at the level of up to 90% of total phenolic content in tubers. Other phenolic acids (caffeic, coumaric, ferulic, and sinapinic) comprise only 10% of the total phenolic acids. Navarre et al. [37] found in their research that the content of chlorogenic acid differed in potatoes with white, yellow, white-and-red, and red-and-purple flesh, and the values were: 31–170; 23–211; 21–231; and 80–473 mg∙100 g^−1^ dry mass, respectively. A number of authors [15,36,37,38] studying potatoes with different skin and flesh colours reported that white-and yellow-fleshed cultivars contain fewer phenolic acids than white-and-red and red-and-purple cultivars. The results are consistent with our own study, which also shows higher phenolic acid content in coloured-fleshed potato cultivars. The same results were also arrived at by Mulinacci et al. [39], whose research demonstrated that the phenolic acid content in purple- and red-fleshed potatoes on average amounted to 101.8 mg∙100 g^−1^ fresh mass; whereas in white-fleshed potatoes, it was only 12.1 mg∙100 g^−1^ fresh mass.

### 3.2. Flavonol Content in Coloured and Bright-Fleshed Potato Tubers [mg 100 g^−1^ FM]

The content of flavonols in the tested potato tubers, both with coloured and bright flesh, averaged 6.62 mg 100 g^−1^ FM. Similar results were obtained in the research by Wierzbicka et al. [20], with an average total flavonol content of 5.82 mg·100 g^−1^ FM. In our own research, we observed the highest flavonol content in coloured-fleshed potatoes (Table 3). Among the flavonoids found in potatoes, quercetin (average quercetin content was 0.29 mg∙100 g^−1^ FM), quercetin-3-O-glucoside (average quercetin content was 0.70 mg∙100 g^−1^ FM), and quercetin-3-O-rutinoside (average quercetin-3-O-rutinoside content was 5.63 mg∙100 g^−1^ FM) were identified. The research results were similar to the ones obtained by Vaitkevičienė et al. [4], which found that in the case of cultivars from an organic farming system, the average content of quercetin was 0.30 mg∙100 g^−1^ FM; quercetin-3-O-glucoside, 0.78 mg 100 g^−1^ FM; and quercetin-3-O-rutinoside, 5.85 mg∙100 g^−1^ FM. The analysis of total flavonol content in potato tubers showed significant differences in their content between cultivars. The content of total flavonols and quercetin-3-O-rutinoside content were also statistically significant in bright- and coloured-fleshed tubers (Table 3).

The highest total flavonol and quercetin-3-O-rutinoside content among the coloured-fleshed cultivars was found in the “Vitelotte” and “Rote Emma” cultivars (8.19 and 8.08; 7.42 and 7.11 mg·100 g^−1^ FM, respectively) (Table 3). The average quercetin content in the cultivars under study was 0.29 mg·100 g^−1^ FM. The highest quercetin content was found in the “Red Emmalie” (0.38 mg·100 g^−1^ FM) and “Rote Emma” cultivars (0.30 mg·100 g^−1^ FM). In the case of bright-fleshed cultivars, the highest quercetin content was observed in the “Tajfun” (0.37 mg·100 g^−1^ FM) and “Bella Rosa” cultivars (0.34 mg·100 g^−1^ FM), whereas the lowest was found in the “Lord” cultivar (0.20 mg·100 g^−1^ FM) (Table 3). A slightly higher average content was arrived at in the research by Wierzbicka et al. [20], amounting to 0.34 mg·100 g^−1^ FM in the cultivars under study and 0.35 mg·100 g^−1^ FM in the case of the “Tajfun” cultivar that they studied.

### 3.3. Anthocyanin Content in Coloured and Bright-Fleshed Potato Tubers [mg 100 g^−1^ FM]

In the research, three groups of anthocyanins were identified: pelargonidin-3,5-di-O-glucoside; peonidin-3,5-di-O-glucoside; and petunidin-3,5-di-O-glucoside. Anthocyanins were not found in the “Lord” or “Tajfun” varieties characterised by white tuber flesh. The predominant pigment was petunidin-3,5-di-O-glucoside, with an average content of (23.15 mg∙100 g^−1^ FM); while the highest value was observed in the “Vitelotte” variety (51.27 mg∙100 g^−1^ FM) (Table 4). Anthocyanin content in the eight potato cultivars under study on average amounted to 25.95 mg 100 g^−1^ FM (Table 4). Our results are consistent with the research by Vaitkevičienė et al. [4], Kita et.al. [40], Tierno et al. [41], and Andre et al. [42]. Their research corroborates that anthocyanins are not present in white- or yellow-fleshed and bright-skinned potatoes, but they can be found in red- and purple-fleshed and/or -skinned tubers. The results of our own research point to the influence of the genetic properties of the cultivars under study on anthocyanin content. This has also been confirmed in research by other authors [4,41,42].

Anthocyanin content was significantly differentiated between the investigated bright-fleshed and -skinned tubers and coloured-fleshed and -skinned tubers. Among the coloured-fleshed cultivars, the highest anthocyanin content was identified in the “Vitelotte” cultivar, and the lowest was identified in “Blue Salad” and “Blue Congo” (Table 4).

### 3.4. Analysis of Total Reducing and Antioxidant Compounds in Coloured and Bright-Fleshed Potato Tubers

The antioxidant activity of the flesh of the coloured (c) and bright (b) potatoes under study was measured by means of antioxidant activity assays: ABTS (cationic radical ABTS^●+^), DPPH (radical DPPH), and FRAP (TPTZ compound). In addition, total polyphenol content (TPC) and total reduced and oxidised ascorbate content were measured. The results of the tests are presented in Table 5 and illustrated in Figure 1. The antioxidant activity of the flesh of the potatoes under study was diverse depending on flesh colour. The FRAP assay indicated higher antioxidant activity of coloured-fleshed potato cultivars. The highest content was identified in the “Vitelotte” cultivar flesh, which was 114% higher than in the “Lord” cultivar. Similar dependencies were found in the case of the DPPH assay; however, in this case, the “Vitelotte” cultivar flesh demonstrated nearly 6.4 times higher antioxidant activity than the “Lord” cultivar. The area of smaller differences in the antioxidant activity of the flesh of the potato cultivars under study was identified by means of the ABTS assay. The highest difference in the antioxidant activity measured in the ABTS assay was found between the flesh of coloured cultivars “Blue Congo” and “Vitelotte” and amounted to 40.8%. The cultivar with the greatest total polyphenol content (TPC) was “Vitelotte”, the flesh of which contained over 300% more polyphenolic compounds than the “Lord” cultivar. However, total polyphenol content (TPC) in the flesh of the remaining coloured cultivars was on average 150% higher than in the “Lord” cultivar. Total ascorbate content ranged from 153.47 µg^−1^ FM in the “Blue Salad” flesh to 417.89 µg^−1^ in the “Vitelotte” flesh, which amounted to an increase by 170% (Table 5). Our results are similar to those obtained by Nemś et al. [43], who showed that the highest antioxidant potential was manifested by the “Blaue Anneliese” cultivar, which is characterised by dark-purple flesh (26.828 μmol Trolox/g DM). According to Hosseini-Beheshti et al. [34] and Rasheed et al. [44], differences in the antioxidant properties of coloured cultivars probably result from not only the total polyphenol content but also differences in anthocyanin stabilisation by other compounds found in tuber dry mass and the different antioxidant potentials of various anthocyanins.

### 3.5. Multidimensional Analysis of Scaled Heat Maps in Coloured and Bright-flesh of Potatoes

The use of the scaled heat map (Figure 1) enabled the grouping of the potato cultivars under study in terms of their antioxidant properties. Data structuring resulted in the identification of clusters of similar properties. Agglomerated research results based on intra-group similarities helped identify five clusters made up of the following groups: clusters 1 and 2, single-element groups, “Blue Salad” and “Vitelotte” cultivars; cluster 3, “Lord” and “Tajfun” cultivars; cluster 4, “Rote Emma” and “Red Emmalie” cultivars; and cluster 5, “Bella Rosa” and “Blue Congo” cultivars.

The conducted multidimensional data mining based on the scaled heat map (Figure 2) in terms of polyphenolic compound content (phenolic acids and flavonoids) and anthocyanin content in eight potato cultivars (three bright-fleshed and five coloured-fleshed) showed inter-cultivar differentiation.

Data agglomeration showed groupings (clusters) characterised by similarities of the analysed compounds. The structure and reduction of the processed data resulted in the identification of four inter-group relations characterised by similar content of polyphenolic compounds and anthocyanins. The first group contained the cultivars with the highest levels of those compounds (“Rote Emma” and “Vitelotte”), whereas the remaining clusters included groups with successively decreasing polyphenolic and anthocyanin content. Cluster 2 contained the “Bella Rosa” and “Red Emmalie” cultivars; cluster 3 contained “Blue Congo” and “Blue Salad”; and cluster 4 contained the cultivars with the lowest levels of those compounds: “Lord” and “Tajfun”.

## 4. Conclusions

Studies of the assessment of tuber flesh of eight potato cultivars showed that the total content of individual phenolic acids varied and depended on the cultivar genotype and the colour of tuber flesh. Studies have shown that potatoes with coloured flesh had a higher content of phenolic acids than those with bright flesh. Moreover, three groups of anthocyanins were identified in potato samples: pelargonidine-3,5-di-O-glucoside, peonidine-3,5-di-O-glucoside, and petunidine-3,5-di-O-glucoside. Anthocyanins were not found in the “Lord” or “Tajfun” cultivars, which are characterised by white flesh of tubers. The antioxidant activity of the investigated potato flesh varied depending on the flesh colour. The FRAP assay showed higher antioxidant activity of potato varieties with coloured flesh. Similar relationships were identified in the case of the DPPH assay. The cultivar with the highest total polyphenol content (TPC) was “Vitelotte”. Summarizing our own research, we can conclude that the varieties of potatoes with coloured flesh are characterised by a higher content of biologically active substances, including phenolic acids, and greater antioxidant properties compared to potato tubers with bright flesh. The consumption of coloured varieties of potatoes can provide more health benefits than traditional bright-flesh varieties. In addition, the research results provide a basis for raising awareness of coloured potato varieties, with the possibility of introducing them into food processing in order to design new food products with a higher level of biologically active ingredients. The results of these studies are expected to raise consumer awareness of the health benefits of coloured-fleshed potato varieties.

## Figures and Tables

**Figure 1 foods-12-00100-f001:**
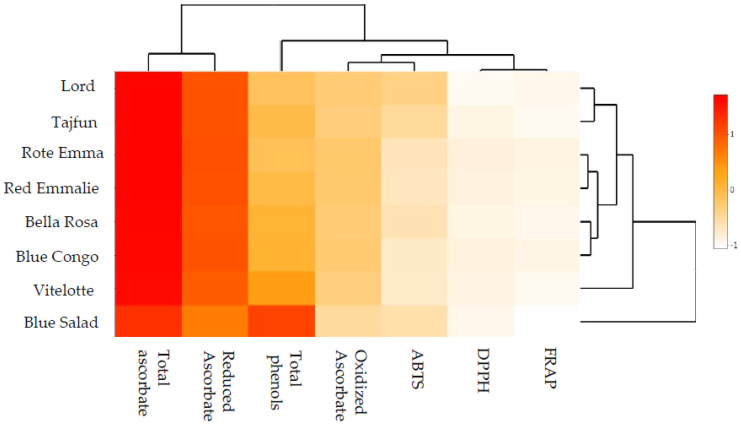
Scaled heat map for potatoes with different skin and flesh colours in terms of antioxidant activity. Darker colour means greater dependence between research factors (higher correlation coefficient). The diagrams on the outside are the reference for cluster analysis.

**Figure 2 foods-12-00100-f002:**
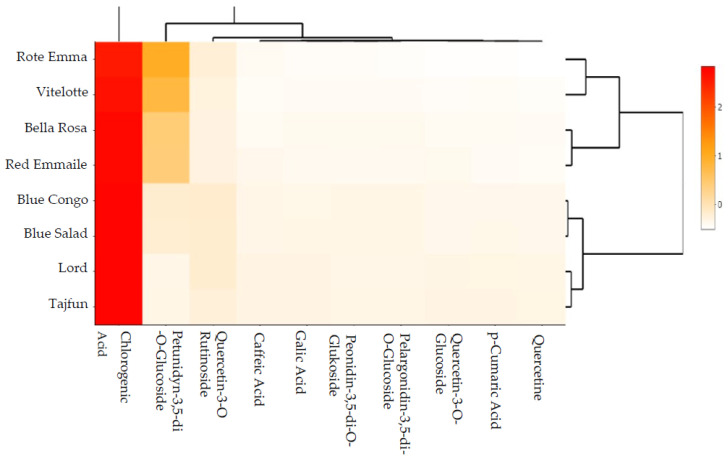
Scaled heat map for potatoes with different skin and flesh colours in terms of polyphenolic and anthocyanin content. Darker colour means greater dependence between research factors (higher correlation coefficient). The diagrams on the outside are the reference for cluster analysis.

**Table 1 foods-12-00100-t001:** Description of potato cultivars.

Cultivar Characteristics	Coloured-fleshed Potatoes	Bright-fleshed Potatoes
“Rote Emma”	“Blue Salad”	“Vitelotte”	“Red Emmalie”	“Blue Congo”	“Bella Rosa”	“Lord”	“Tajfun”
Maturity	MediumEarly	Early	Early	MediumEarly	Late	Very early	Early	Medium early
Flesh colour	Dark red	Light blue-purple	DarkPurple	Red	Violet	Yellow	Light yellow	LightYellow
Skin colour	Red	Dark blue-purple	DarkPurple	Red	Violet	Pink	Light yellow	Light yellow
Tuber shape	Long oval	Shortoval	Long	Long	Oval	Roundoval	Round oval	Oval
Cooking type	AB	A	AB	AB	BC	AB	AB	B-BC
Picture	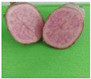	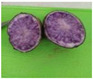	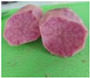	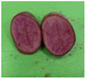	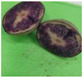	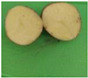	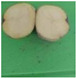	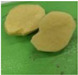

Cooking type A: Potatoes do not overcook. Their flesh remains firm after cooking and can be cut easily (European Association for Potato Research cooking type scale); Cooking type B: Potatoes that are slightly floury, disintegrating, and with rather fine-textured flesh (European Association for Potato Research cooking type scale); Cooking type C: Potatoes that are dry, with floury flesh, coarse, and a rather marked disintegration (European Association for Potato Research cooking type scale) [4].

**Table 2 foods-12-00100-t002:** Phenolic acid content in coloured and bright-fleshed potato tubers (mg 100 g^−1^ FM).

Cultivar	Phenolic Acids (sum)	Chlorogenic Acid	Gallic Acid	p-Coumaric Acid	Caffeic Acid
mg∙100 g^−1^ fresh mass (FM) ± SD					
^a^ Rote Emma (c)	80.49 ± 0.10 ^b–h^	75.60 ± 0.32 ^b–g,h^	1.55 ± 0.09 ^e,f,g,h^	0.79 ± 0.03 ^e,f,g,h^	2.55 ± 0.08 ^b,c,d,e,f,g,h^
^b^ Blue Salad (c)	120.62 ± 0.70 ^a,c–h^	117.71 ± 0.27 ^a,c–h,^	1.37 ± 0.06 ^e,g,h^	0.71 ± 0.06 ^f,g,h^	0.83 ± 0.08 ^a,d,g,h^
^c^ Vitelotte (c)	129.94 ± 0.69 ^a,b,d–h^	126.77 ± 0.55 ^a,b,d–h^	1.44 ± 0.10 ^e,g,h^	0.80 ± 0.06 ^f,g,h^	0.93 ± 0.04 ^a,d,f,g,h^
^d^ Red Emmalie (c)	77.85 ± 0.48 ^a,b,c,e–h^	73.37 ± 0.75 ^a,b,c,e–h^	1.42 ± 0.11 ^e,g,h^	0.73 ± 0.02 ^e,f,g,h^	2.33 ± 0.05 ^a,b,c,e,f,g,h^
^e^ Blue Congo (c)	107.96 ± 0.43 ^a–d,f,g,h^	105.55 ± 0.66 ^a–d,f,g,h^	0.88 ± 0.05 ^a,b,c,d,f^	0.58 ± 0.06 ^a,c,d^	0.95 ± 0.07 ^a,d,f,g,h^
^f^ Bella Rosa (b)	66.50 ± 0.73 ^a–e^	64.11 ± 0.75 ^a–e^	1.25 ± 0.03 ^a,e,g,h,f^	0.48 ± 0.03 ^a,b,c,d^	0.66 ± 0.05 ^a,c,d,e^
^g^ Lord (b)	64.78 ± 0.55 ^a–e^	62.95 ± 0.23 ^a–e^	0.69 ± 0.04 ^a,b,c,d,f^	0.56 ± 0.06 ^a,b,c,d^	0.58 ± 0.05 ^a,b,c,d,e^
^h^ Tajfun (b)	65.30 ± 0.91 ^a–e^	63.46 ± 0.69 ^a–e^	0.72 ± 0.09 ^a,b,c,d,f^	0.51 ± 0.05 ^a,b,c,d^	0.63 ± 0.05 ^a,b,c,d,e^
Means	89.19	86.19	1.18	0.64	1.18

Legend: a—Rote Emma (c); b—Blue Salad (c); c—Vitelotte (c); d—Red Emmalie (c); e—Blue Congo (c); f—Bella Rosa (b); g—Lord (b); h—Tajfun (b); (c)—coloured flesh, (b)—bright flesh, respectively; and SD—standard deviation. The digits at the values stand for statistically significant differences between means tested according to ANOVA in Tukey’s post-hoc test.

**Table 3 foods-12-00100-t003:** Flavonol content in coloured and bright-fleshed potato tubers (mg 100 g^−1^ FM).

Cultivar	Flavonols (sum)	Quercetin	Quercetin-3-O-Glucoside	Quercetin-3-O-Rutinoside
mg 100 g^−1^ fresh mass (FM) ± SD				
^a^ Rote Emma (c)	8.08 ± 0.04 ^d,f,g,h^	0.30 ± 0.03 ^b,g^	0.67 ± 0,04 ^b,d,g^	7.11 ± 0.10 ^d,f,g,h^
^b^ Blue Salad (c)	8.00 ± 0.12 ^d,f,g,h^	0.19 ± 0.02 ^a,d,f,g^	0.49 ± 0.04 ^a,d^	7.32 ± 0.13 ^d,f,g,h,^
^c^ Vitelotte (c)	8.19 ± 0.16 ^d,f,g,h^	0.25 ± 0.03 ^b,g^	0.53 ± 0.04 ^d,g^	7.41 ± 0.13 ^d–h^
^d^ Red Emmalie (c)	6.10 ± 0.08 ^a,b,c,e–h^	0.38 ± 0.08 ^b,g^	1.79 ± 0.14 ^a,b,c,e–h^	3.93 ± 0.21 ^a,b,c,e,f,h^
^e^ Blue Congo (c)	7.80 ± 0.09 ^d,f,g,h^	0.31 ± 0.05 ^b,g^	0.55 ± 0.04 ^d,g^	6.94 ± 0.21 ^a,b,c,e,f–h^
^f^ Bella Rosa (b)	4.06 ± 0.07 ^a–e,h^	0.34 ± 0.04 ^b,c,g^	0.64 ± 0.05 ^d,g^	3.06 ± 0.07 ^a,b,d,f–h^
^g^ Lord (b)	3.82 ± 0.09 ^a–h^	0.20 ± 0.04 ^a,d,e,h^	0.33 ± 0.04 ^a,c–f,h^	3.29 ± 0.07 ^a–e,h^
^h^ Tajfun (b)	2.97 ± 0.09 ^a–g^	0.37 ± 0.04 ^b,c,g^	0.63 ± 0.04 ^d,g^	1.97 ± 0.02 ^a–g^
Means	6.62	0.29	0.70	5.63

Legend: a—Rote Emma (c); b—Blue Salad (c); c—Vitelotte (c); d—Red Emmalie (c); e—Blue Congo (c); f—Bella Rosa (b); g—Lord (b); h—Tajfun (b); (c)—coloured flesh, (b)—bright flesh, respectively; and SD—standard deviation. The digits at the values stand for statistically significant differences between means tested according to ANOVA in Tukey’s post-hoc test.

**Table 4 foods-12-00100-t004:** Anthocyanin content in coloured- and bright-fleshed potato tubers (mg 100 g^−1^ FM).

Cultivar	Anthocyanins (sum)	Petunidin-3,5-di-O-Glucoside	Pelargonidin-3,5-di-O-Glucoside	Peonidin-3,5-di-O-Glucoside
mg 100 g^−1^ fresh mass (FM) ± SD				
^a^ Rote Emma (c)	39.99 ^b–f^	37.22 ^b–h^	1.34 *	1.43 *
^b^ Blue Salad (c)	9.40 ^a,c,d,f^	6.55 ^a,c–e,g,h^	1.41 *	1.44 *
^c^ Vitelotte (c)	54.09 ^a,b,d–f^	51.27 ^a,b,d,e,f^	1.37 *	1.45 *
^d^ Red Emmalie (c)	22.57 ^a–c,e,f^	19.76 ^a,b,c,e,f^	1.39 *	1.42 *
^e^ Blue Congo (c)	9.54 ^a,c,d,f^	6.67 ^a,c,d,f^	1.42 *	1.45 *
^f^ Bella Rosa (b)	20.15 ^a–e^	17.42 ^a–e^	1.32 *	1.41 *
^g^ Lord (b)	0.00	0.00	0.00	0.00
^h^ Tajfun (b)	0.00	0.00	0.00	0.00
Means	25.95	23.15	1.37	1.43

Legend: a—Rote Emma (c); b—Blue Salad (c); c—Vitelotte (c); d—Red Emmalie (c); e—Blue Congo (c); f—Bella Rosa (b); g—Lord (b); h—Tajfun (b); * no statistically significant differences; and (c)—coloured flesh, (b)—bright flesh, respectively. The digits at the values stand for statistically significant differences between means tested according to ANOVA in Tukey’s post-hoc test.

**Table 5 foods-12-00100-t005:** Total, reduced, and oxidised ascorbate contents and antioxidant properties as determined in FRAP, ABTS, and DPPH assays.

Cultivar	FRAP (μmol Trolox ∗ g^−1^ DW) ± SD	ABTS (μmol Trolox ∗ g^−1^ DW) ± SD	DPPH (μmol TE ∗ g^−1^ DW) ± SD	Total Phenolic Compounds (mg GAE ∗ 100 g^−1^ FM) ± SD	Total Ascorbate Content (μg ∗ g^−1^ FM) ± SD	Reduced Ascorbate Content (μg ∗ g^−1^ FM) ± SD	Oxidised Ascorbate Content (μg ∗ g^−1^ FM) ± SD
^a^ Rote Emma (c)	3.74 ± 0.18 ^b,c^	34.46 ± 0.28 ^c–h^	9.21 ± 0.16 ^b,c,e,f,g^	111.92 ± 0.42 ^b–h^	368.55 ± 0.88 ^b–h^	272.73 ± 0.61 ^b–h^	95.82 ± 0.39 ^b–h^
^b^ Blue Salad (c)	5.63 ± 0.24 ^a,c–h^	34.88 ± 0.35 ^c–h^	12.77 ± 0.29 ^a,c–h^	143.93 ± 0.43 ^a,c–h^	153.47 ± 0.38 ^a,c–h^	113.57 ± 0.54 ^a,c–h^	39.90 ± 0.47 ^a,c–h^
^c^ Vitelotte (c)	6.89 ± 0.26 ^a,b,d–h^	40.93 ± 0.40 ^a,b,d–h^	18.30 ± 0.15 ^a,b,d–h^	219.16 ± 1.03 ^a,b,d–h^	417.83 ± 0.52 ^a,b,d–h^	309.19 ± 0.14 ^a,b,d–h^	108.64 ± 0.81 ^a,b,d–h^
^d^ Red Emmalie (c)	3.87 ± 0.52 ^b,c^	37.22 ± 0.46 ^a,b,c,e,h^	9.33 ± 0.30 ^b,c,e–h^	139.53 ± 0.83 ^a,b,c,e–h^	407.92 ± 1.04 ^a,b,c,e–h^	301.86 ± 0.50 ^a,b,c,e–h^	106.06 ± 0.52 ^a,b,c,f,g,h^
^e^ Blue Congo (c)	3.36 ± 0.63 ^b,c^	29.06 ± 0.30 ^a–h^	11.72 ± 0.20 ^a–d,f,g,h^	155.72 ± 0.64 ^a–d,f,g,h^	401.12 ± 1.72 ^a–d,f,g,h^	296.83 ± 0.76 ^a–d,f,g,h^	104.29 ± 1.07 ^a,b,c,g,h^
^f^ Bella Rosa (b)	3.67 ± 0.17 ^b,c^	36.17 ± 0.51 ^a,b,c,e,g,h^	8.17 ± 0.35 ^a–e,g^	107.82 ± 0.82 ^a–e,g,h^	274.34 ± 1.22 ^a–e,g,h^	203.01 ± 1.31 ^a–e,g,h^	71.33 ± 1.05 ^a–e,g,h^
^g^ Lord (b)	3.22 ± 0.46 ^b,c^	38.39 ± 0.38 ^a,b,c,e,g,h^	3.27 ± 0.03 ^a–e,f^	52.99 ± 0.65 ^a–e,f,h^	169.22 ± 1.11 ^a–f,h^	125.22 ± 1.12 ^a–f,h^	44.00 ± 0.69 ^a–f,h^
^h^ Tajfun (b)	3.41 ± 0.55 ^b,c^	39.12 ± 0.55 ^a–f^	8.61 ± 0.31 ^b–e,g^	71.27 ± 0.66 ^a–g^	198.25 ± 1.31 ^a–g^	146.71 ± 0.97 ^a–g^	51.54 ± 0.82 ^a–g^

Legend: a—Rote Emma (c); b—Blue Salad (c); c—Vitelotte (c); d—Red Emmalie (c); e—Blue Congo (c); f—Bella Rosa (b); g—Lord (b); h—Tajfun (b); (c)—coloured flesh, (b)—bright flesh, respectively; and SD—standard deviation. The digits at the values stand for statistically significant differences between means tested according to ANOVA in Tukey’s post-hoc test.

## Data Availability

Data is contained within the article.

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
