# Peer review of "Phenolic Acid Content and Antioxidant Properties of Edible Potato (*Solanum tuberosum* L.) with Various Tuber Flesh Colours"

_foods, 2022, doi:10.3390/foods12010100_

Round 1

Reviewer 1 Report

Dear Authors,

The presented study addresses the variability of bioactive compounds in potato tubers depending on the variety and color of the flesh.

Bioactive compounds in plant materials are still a current topic.Presented work is interesting.

Bioactive compounds in potato tubers have been studied by other authors. However, this research can contribute to the selection of the varieties, since it is important to identify the varieties with higher content of bioactive substances.

This research confirms the findings of other authors.

The paper is weel writen.

The text is clear and easy to read.

The conclusions are consistent with the evidence and arguments presented.

The main question, whether a variety has an impact on the content of bioactive substances, has been answered. However, other influences should be discussed, such as locality, and maturity stage. Another important factor is the processing of potatoes, during which the decrease of bioactive substances occurs.

presented work is well written, I would only recommend discussing  other influences e.g. locality, or maturity of plant material   (for example this work https://doi.org/10.3390/agronomy12061454 )

Author Response

Response to Reviewer 1 Comments

Manuscript ID: foods-1997809

Title: Phenolic acid content and antioxidant properties of edible potato (Solanum tuberosum L.) with various tuber flesh colours

Dear Reviewer 1,

Thank you very much for your in-depth review of our manuscript. Thank you for your appreciation for our manuscript. Thank you for the proposal to discuss the impact of location, maturation steps or processing effects on bioactive substances content. These are valuable comments and require several years of research. Currently, we are after one year of research. After three years of research, we will prepare such a manuscript for publication.

Thank you very much for the literature suggestion https://doi.org/10.3390/agronomy12061454). We've included it in our manuscript ,, Franková, H .; Musilová, J .; Arvay, J .; Harangozo, L .; Šnirc, M .; Vollmannová, A .; Lidiková, J .; Hegedűsová, A .; Jaško, E. Variability of bioactive substances in potatoes (Solanum Tuberosum L.) depending on the variety and maturity. Agronomy 2022, 12, 1454. https://doi.org/10.3390/agronomy12061454 ''

The comments and research suggestions introduced into the text will significantly increase its scientific value. The text has been corrected in line with the reviewers' comments. We send the corrected work in the attachment. Corrections are marked in yellow.

We hope that the manuscript in its current form meets the requirements of the Foods journal.

Reviewer 2 Report

There are too many relevant articles, and the research content is similar to some of the relevant articles. This article has no originality, and can not meet the quality requirements of this Journal.

Author Response

Response to Reviewer 2 Comments

Manuscript ID: foods-1997809

Title: Phenolic acid content and antioxidant properties of edible potato (Solanum tuberosum L.) with various tuber flesh colours

Dear Reviewer 2, Thank you very much for your insightful review of our manuscript. We have corrected the text according to the comments of other reviewers. We have revised and completed:

  • abstract,
  • introduction,
  • research material and methods,
  • results and discussion,
  • conclusions,
  • We corrected the description of the tables.
  • We have revised the purpose of the study.
  • We have made linguistic and grammatical corrections to the entire manuscript.

We send the corrected paper in the appendix. The corrections are marked in yellow.

We very much ask the reviewer to read and acknowledge our corrections again. We also ask that you recognize our research and results as research that is important and needed by today's consumer. We believe that bioactive compounds in plant materials is still a timely topic, and knowledge of our research results will increase consumer awareness of the health benefits of colorful potato varieties.

We request the reviewer's approval of revisions to the text and permission to publish our manuscript in the journal Foods.

Reviewer 3 Report

The manuscript “Phenolic acid content and antioxidant properties of edible potato (Solanum tuberosum L.) with various tuber flesh colours” deals with the estimation of the phenolic content of about eight coloured potato cultivars. Although the conceptualization of the study is not new, and many reports on similar research are already there in the public domain. However, the data generated will be helpful in designing further experiments for breeding nutritionally superior potato cultivars. The author needs to address the following points.

Comment

·       Abstraction section: No need to provide the latitude and longitude here. Kindly provide in the material and method section.

·       LN 27 and 29: Please make the unit in uniform format.

·       The abstract section needs to be improved significantly.

·       LN 57-63: Please rewrite the line.

·       LN 70-72: The line is not clear. Please rephrase the line.

·       The objective of the research should be redefined.

·       The introduction section is too short. This need to be revisited. The author must include the importance of phenolic compounds and their mechanism in preventing chronic diseases and cancer.

·       LN 122: Please be consistent with the use of the format of the unit mentioned throughout the manuscript.

·       Table 2: The author can use different alphabetical characters instead of the number to define Tukey’s post-hoc test. Similarly, check for other tables too.  

·       The caption of Figures 1 and 2 need to be explained in more detail. 

Author Response

Response to Reviewer 3 Comments

Manuscript ID: foods-1997809

Title: Phenolic acid content and antioxidant properties of edible potato (Solanum tuberosum L.) with various tuber flesh colours

Dear Reviewer 3,

We are very grateful for the insightful review of our manuscript. The reviewers remarks and suggestions, which have been implemented into the text, will significantly increase the scientific value of it. The text has been improved according to the reviewers' remarks, and below we have enclosed comments on the revised version. We hope that the manuscript in its present form meets the requirements of the Foods journal.

Point 1. Abstraction section: No need to provide the latitude and longitude here. Kindly provide in the material and method section.

Response 1: Thank you for the remark, corrected to:

Abstract: ,,The raw material used in this study included tubers of 8 edible potato cultivars. Five cultivars were potato tubers with coloured flesh: “Rote Emma”, “Blue Salad”, “Vitelotte”, “Red Emmalie”, “Blue Congo”, and three with bright flesh: “Bella Rosa”, “Lord”, “Tajfun”. The edible potato tubers were obtained from the plantation of the Carpathian State College in Krosno (latitude 21°46’E; longitude 49°42’N). The study was conducted during the vegetation period from 2020 to 2021’’.

In abstraction section, the information regarding the geographical location of the test site was removed.

The correct notation is:

Abstract: ,,The purpose of the paper was to assess the phenolic acid content and antioxidant properties in potato cultivars with coloured flesh and bright flesh. The study material included 8 cultivars of edible potato grown in temperate climate in Poland. Five cultivars were potato tubers with coloured flesh: “Rote Emma”, “Blue Salad”, “Vitelotte”, “Red Emmalie”, “Blue Congo”, and three with bright flesh: “Bella Rosa”, “Lord”, “Tajfun”. In all potato samples under study, 4 phenolic acids were identified……’’

Point 2. LN 27 and 29: Please make the unit in uniform format.

Response 2: Thank you for the remark, corrected and corrected and uniform all units in the abstract and throughout the manuscript.

Point 3. The abstract section needs to be improved significantly.

Response 3: Thank you for the remark, the entire summary has been corrected and completed. Its corrected form is

Abstract the incorrect form:

Abstract: The raw material used in this study included tubers of 8 edible potato cultivars. Five cultivars were potato tubers with coloured flesh: “Rote Emma”, “Blue Salad”, “Vitelotte”, “Red Emmalie”, “Blue Congo”, and three with bright flesh: “Bella Rosa”, “Lord”, “Tajfun”. The edible potato tubers were obtained from the plantation of the Carpathian State College in Krosno (latitude 21°46’E; longitude 49°42’N). The study was conducted during the vegetation period from 2020 to 2021. The study focused on the content of: phenolic acids, total and reduced ascorbate, flavonols, anthocyanins, and the antioxidant activity tested by means of ABTS, DPPH, and FRAP assays. The highest levels of phenolic acids were found in coloured-fleshed tubers, and the lowest in bright-fleshed tubers. The predominant acid in all samples under study was chlorogenic acid, whose level ranged from 63.46 mg.100 g-1 to 126.77 mg.100 g-1 FM. The highest phenolic acid content among the coloured-fleshed tubers was reported in the “Vitelotte” and “Blue Salad” cultivars (129.94 and 120.62 mg.100-1 FM, respectively). According to a quantitative analysis, flavonol content in the potato tubers, with both coloured and bright flesh, on average amounted to 6.62 mg 100 g-1 FM. The identification of anthocyanins in the qualitative and quantitative potato analysis pointed to the presence of those compounds in coloured-fleshed cultivars. Anthocyanins were not identified in bright-fleshed potato cultivars (with the exception of “Bella Rosa” characterised by white flesh and pink skin). The measured anthocyanins were: pelargonidin-3,5-di-O-glucoside, peonidin-3,5-di-O-glucoside, petunidin-3,5-di-O-glucoside. The analysis of total reducing and antioxidant compounds in tubers of the cultivars under study indicated significant differences in these compounds between cultivars and flesh colour. Coloured-fleshed cultivars demonstrated significantly higher levels than bright-fleshed tubers. In conclusion of our own research, we can state that coloured-fleshed tubers are characterised by higher contents of biologically active substances as compared to bright-fleshed potato tubers. The consumption of coloured-fleshed potato cultivars can potentially have more health benefits than traditional cultivars.

Its corrected form is

Abstract: The purpose of the paper was to assess the phenolic acid content and antioxidant properties in potato cultivars with coloured flesh and bright flesh. The study material included 8 cultivars of edible potato grown in temperate climate in Poland. Five cultivars were potato tubers with coloured flesh: “Rote Emma”, “Blue Salad”, “Vitelotte”, “Red Emmalie”, “Blue Congo”, and three with bright flesh: “Bella Rosa”, “Lord”, “Tajfun”. In all potato samples under study, 4 phenolic acids were identified: chlorogenic acid, gallic acid, p-coumaric acid, caffeic acid. The predominant acid was chlorogenic acid, whose levels ranged 62.95 mg.100 g-1 FM to 126.77 mg.100 g-1 FM. Total content of the individual identified phenolic acids was diversified and depended on the genotype of the cultivar and the tuber flesh colour, with coloured-fleshed potatoes having higher phenolic acid content as compared to bright-fleshed potato cultivars. The average concentration of phenolic acids in the samples was 89.19 mg∙100 g-1 FM, and for the individual phenolic acids identified were as follows: chlorogenic acid (86.19 mg∙100 g-1 FM), gallic acid (1.18 mg∙100 g-1 FM), p-coumaric acid (0.64 mg∙100 g-1 FM), caffeic acid (1.18 mg∙100 g-1 FM). In addition, 3 groups of anthocyanins were identified: pelargonidin-3,5-di-O-glucoside, peonidin-3,5-di-O-glucoside, petunidin-3,5-di-O-glucoside. Anthocyanins were not found in 'Lord' and 'Tajfun' varieties characterized by white tuber flesh. The dominant pigment was petunidin-3,5-di-O-glucoside, the average content of which was (23.15 mg∙100 g-1 FM), while the highest value was observed in the 'Vitelotte' variety (51.27 mg∙100 g-1 FM). The antioxidant activity of the flesh of the potatoes under study was diversified depending on flesh colour. The FRAP assay indicated higher antioxidant activity of coloured-fleshed potato cultivars. The highest content was identified in the “Vitelotte” cultivar flesh, which was 114% higher than in the “Lord” cultivar. Similar dependencies were found in the case of the DPPH assay; however, in this case, the “Vitelotte” cultivar flesh demonstrated nearly 6.4 times higher antioxidant activity than the “Lord” cultivar. Summarizing our own research, it can be concluded that potato varieties with colored flesh are characterized by a higher content of biologically active substances, including phenolic acids, and antioxidant properties compared to potato tubers with light flesh.

Point 4. LN 57-63: Please rewrite the line.

Response 4: Thank you for the remark. Removed line 57-63.

the incorrect form:

,,The health benefits of coloured-fleshed and -skinned potatoes have been confirmed in the research by Akyol et al. [8], Kazimierczak et al. [9], and Liao et al. [10], who identified phenolic acids, mainly chlorogenic acid, and to a lesser extent, caffeic, cinnamic, p-coumaric, ferulic, and sinapinic acid, as well as flavonoids, mainly catechin and epicatechin, According to Brown et al. [11] and Lachman et al. [6], coloured-fleshed and -skinned potato cultivars also contain flavonol aglycones, the main group of plant phenols that are potent antioxidants. The authors believe that red-fleshed potatoes contain acylated glycosides of pelargonidin, whereas purple-fleshed potatoes contain acylated glycosides of malvidin, petunidin, peonidin, and delphinidin. These compounds are responsible for the antioxidant properties of potatoes.’’

Its corrected form is:

According to Zhang et al. [14] and Koszowska et al. [11], a diet rich in antioxidants is beneficial to health as it decreases the incidence of cardiovascular diseases, diabetes, cancers, and osteoporosis. Research by Brown et al. [15] and Lachman et al. [6] showed that coloured-fleshed and -skinned potato cultivars also contain flavonol aglycones, the main group of plant phenols that are potent antioxidants. These compounds are responsible for the antioxidant properties of potatoes. As compared to white- or yellow-fleshed cultivars, coloured-fleshed and -skinned potatoes contain almost three times as many polyphenolic compounds, including anthocyanins, not present in traditional cultivars. The authors believe that red-fleshed potatoes contain acylated glycosides of pelargonidin, whereas purple-fleshed potatoes contain acylated glycosides of malvidin, petunidin, peonidin, and delphinidin.

Point 5. LN 70-72: The line is not clear. Please rephrase the line.

Response 5: Thank you for the remark.

the incorrect form:

,,Furthermore, the in vivo test conducted by Mulero et al. [13] showed high antioxidant activity of purple potato flakes, which involved the inhibition of RNA and linolenic acid oxidation. They have also shown that the antioxidant properties of purple potato flakes enhance hepatic Mn-SOD, Cu/Zn-SOD and GSH-Px mRNA expression.’’

Its corrected form is:

,,In addition, a study by Han et al [19], on rats fed purple potato flakes showed that the flakes have antioxidant functions with respect to radical scavenging and inhibition of linoleic acid oxidation, and improved antioxidant potential in rats by increasing hepatic mRNA expression of Mn-SOD, Cu/Zn-SOD and GHS-Px.’’

Point 6. The objective of the research should be redefined.

Response 6: Thank you for the remark.

The aim of the work has been changed to (changes are marked in red):

,,The introduction of coloured fleshed potatoes into food processing can improve the potato product diversity, which will stand out with not only their colour but also higher content of biologically active compounds. Therefore, the consumption of coloured-fleshed potato cultivars can potentially have more health benefits than traditional white-fleshed cultivars. It can have health benefits related to antioxidants, such as anti-cancer, anti-ageing, and anti-inflammatory properties. Source literature reports that coloured-fleshed potatoes are beneficial to health. However, there has been little research on the influence of potato cultivars on the phenolic acid content and antioxidant properties of potato cultivars with coloured flesh and bright flesh. Therefore, the aim of this paper was to gain such knowledge and raise consumer awareness of the health benefits of coloured potato cultivars.’’

Point 7. The introduction section is too short. This need to be revisited. The author must include the importance of phenolic compounds and their mechanism in preventing chronic diseases and cancer.

Response 7: Thank you for the remark.

Point 8. LN 122: Please be consistent with the use of the format of the unit mentioned throughout the manuscript.

Response 8: Thank you for the remark. The reviewer's comments have been corrected, incorrectly written units have been corrected throughout the text

Point 9.Table 2: The author can use different alphabetical characters instead of the number to define Tukey’s post-hoc test. Similarly, check for other tables too.  

Response 9: Thank you for the remark. The introduction to the work was supplemented with the importance of phenolic compounds and their mechanism of action in the prevention of chronic and neoplastic diseases.  The correct text is:

,,Potato (Solanum tuberosum L.) is one of the most popular foods consumed around the world, after rice and wheat [1]. Its nutritional value is based on the high content of carbohydrates, mainly starch, and high-quality protein, macro- and micro-elements and bioactive ingredients [2, 3]. Scientists studying potato quality have paid particular attention to coloured-skinned and coloured-fleshed potatoes, which are still little-known and not very popular raw materials in terms of use in households and food processing [4, 5]. Within the identified genetic materials of coloured-fleshed and -skinned potatoes, chemical compounds were found that protect human cells from damage caused by free radicals, prevent oxidized low-density lipoprotein cholesterol, and contribute to lower incidence of some types of cancer, neurodegenerative diseases, osteoporosis, and diabetes [6, 7]. The health benefits of coloured-fleshed and -skinned potatoes were confirmed in the research by Akyol et al. [8], Kazimierczak et al. [9], and Liao et al. [10], who identified polyphenols in potato, most of which were phenolic acids, mainly chlorogenic acid, and to a lesser extent, caffeic, cinnamic, p-coumaric, ferulic, and sinapinic acid, as well as flavonoids, mainly catechin and epicatechin. The presence of polyphenols in human diet is crucial in the prevention of a number of lifestyle diseases [4, 11, 12, 13, 14]. According to those authors, polyphenolic compounds in the diet can help maintain good body condition, prevent many diseases, and facilitate the treatment of existing disorders. According to Zhang et al. [14] and Koszowska et al. [11], a diet rich in antioxidants is beneficial to health as it decreases the incidence of cardiovascular diseases, diabetes, cancers, and osteoporosis. Research by Brown et al. [15] and Lachman et al. [6] showed that coloured-fleshed and -skinned potato cultivars also contain flavonol aglycones, the main group of plant phenols that are potent antioxidants. These compounds are responsible for the antioxidant properties of potatoes. As compared to white- or yellow-fleshed cultivars, coloured-fleshed and -skinned potatoes contain almost three times as many polyphenolic compounds, including anthocyanins, not present in traditional cultivars. The authors believe that red-fleshed potatoes contain acylated glycosides of pelargonidin, whereas purple-fleshed potatoes contain acylated glycosides of malvidin, petunidin, peonidin, and delphinidin. In their research, Piątkowska et al. [16] reported that products containing anthocyanin compounds are beneficial to health and have anti-atherosclerotic, anti-inflammatory, antioxidant and anti-cancer properties. According to Zhang et al. [14] and Koszowska et al. [11], anthocyanins have the ability to prevent brittleness of blood vessels and capillaries, and can stimulate rhodopsin production necessary in the vision process. In addition, they have a positive influence on decelerating the oxidation of LDL cholesterol, which makes up atheromatous plaque. The health benefits of anthocyanins was also confirmed in the research by Zawistowski et al. [17], and Zhang et al. [14], who reported that anthocyanins not only have protective properties in case of neoplastic changes, but also have a positive effect on the lipid profile of the system, resulting in its decrease, as well as increase insulin sensitivity. The research by Jiang et al. [18] showed that anthocyanins present in purple-fleshed potatoes participate in liver regeneration following alcohol-related damage. The multi-directional anthocyanin activity in the system gives hope for their preventive or medicinal use in the treatment of many diseases [4, 12, 13, 14, 18]. In addition, a study by Han et al [19], on rats fed purple potato flakes showed that the flakes have antioxidant functions with respect to radical scavenging and inhibition of linoleic acid oxidation, and improved antioxidant potential in rats by increasing hepatic mRNA expression of Mn-SOD, Cu/Zn-SOD and GHS-Px. Other authors’ studies have shown that coloured-fleshed potatoes demonstrate high antioxidant activity and have the potential to reduce oxidative stress [1, 11, 15, 20, 23, 24]. Coloured-fleshed potatoes are recommended for consumption and production of fried and dried products, in particular due to good organoleptic characteristics (flavour, smell, texture, and colour), but also higher content of biologically active ingredients [4, 20]. The introduction of coloured fleshed potatoes into food processing can improve the potato product diversity, which will stand out with not only their colour but also higher content of biologically active compounds. Therefore, the consumption of coloured-fleshed potato cultivars can potentially have more health benefits than traditional white-fleshed cultivars. It can have health benefits related to antioxidants, such as anti-cancer, anti-ageing, and anti-inflammatory properties. Source literature reports that coloured-fleshed potatoes are beneficial to health. However, there has been little research on the influence of potato cultivars on the phenolic acid content and antioxidant properties of potato cultivars with coloured flesh and bright flesh. Therefore, the aim of this paper was to gain such knowledge and raise consumer awareness of the health benefits of coloured potato cultivars’’.

Point 10. The caption of Figures 1 and 2 need to be explained in more detail. 

Response10: Thank you for the remark. Corrections were made in accordance with the comments of the reviewer. Caption under figures 1 and 2 explained in more detail.

 Figure 1. Scaled heat map for potatoes with different skin and flesh colours in terms of antioxidant activity

Figure 2. Scaled heat map for potatoes with different skin and flesh colours in terms of polyphenolic and anthocyanin content

The correct text is:

Figure 1. Scaled heat map for potatoes with different skin and flesh colours in terms of antioxidant activity. Darker color means greater dependence between research factors (higher correlation coefficient). The diagrams on the outside are the reference for cluster analysis.

Figure 2. Scaled heat map for potatoes with different skin and flesh colours in terms of polyphenolic and anthocyanin content.

We would like to thank the reviewer for his kind comments and ask him to accept the corrections in the text.

Reviewer 4 Report

 Cebulak et al reported the bioactive and antioxidant activities of different potato cultivars. This not a new study but the current results contribute to the growing literature on the beneficial effects of potato on humans. The authors need improve their assay descriptions as well as the results and discussion as the current form is difficult to read couple with grammar issue throughout the manuscript. The authors should seek English editing service before resubmitting the revise on for consideration. Below are some comments to help improve the manuscript.  

Line 18-23 irrelevant details in abstract. Please revise and summarize vital information here

Line 23 content of: phenolic????

Line 23-24 “The study focused on the content of: phenolic acids, total and reduced ascorbate, flavonols, 23 anthocyanins, and the antioxidant activity tested by means of ABTS, DPPH, and FRAP assays” does flow well. Also, no abbreviation in abstract

Line 25 “highest levels of phenolic acids”…..please include range of this

Line 26 please consider deleting “under study”

Line 28 “was reported” ….please remove this

Line 29 please remove this “According to a quantitative analysis”

Line 34 please consider adapting “Pelargonidin-3,5-di-O-glucoside, peonidin-3,5-di-O-34 glucoside, petunidin-3,5-di-O-glucoside were the anthocyanins detected.”

Line 40-41 potentially improve health benefits????

Line 59-61 please correct the grammar

Line 70 how is “inhibition of RNA and linolenic acid oxidation” related to antioxidant activity? Please revise

Line 75 “and smell substances”????

Line 74-76 please revise

Line 77 “potato products offer” what do mean?

76-88 please improve.

Line 95 please include the soil and climatic characteristics of the area where the potatoes were cultivated. Also were they treated after harvesting before storage prior to the study? These are vital information to include.

Line 104 delete “For the study” and join the sentence with the preceding one. Also “slices (approx. 10 mm)” with what?  

Line 111 delete “means of”

Line 112 should be “as previously described [9].”

Line 113 -122 poorly described please improve

Line 124-226 needs further improvements to enhance the descriptions

Line 234 research objects???

Table 2 where are the SD OR SEM not reported. Also what is the essence of those superscript numbers for? Doesn’t make the table nice and self explanatory. Please improve all tables in the manuscript

Results and discussions need improvements. Difficult to read the current form.

Author Response

Response to Reviewer 4 Comments

Manuscript ID: foods-1997809

Title: Phenolic acid content and antioxidant properties of edible potato (Solanum tuberosum L.) with various tuber flesh colours

Dear Reviewer,

We are very grateful for the insightful review of our manuscript. The reviewers remarks and suggestions, which have been implemented into the text, will significantly increase the scientific value of it. The text has been improved according to the reviewers' remarks, and below we have enclosed comments on the revised version. We hope that the manuscript in its present form meets the requirements of the Foods journal.

Point 1. Line 18-23 irrelevant details in abstract. Please revise and summarize vital information here

Response 1: Thank you for the remark, the content in the abstract of lines 18-23 has been corrected. Redundant information on the place of testing has been removedThe correct content is:

Abstract: The raw material used in this study included tubers of 8 edible potato cultivars. Five cultivars were potato tubers with coloured flesh: “Rote Emma”, “Blue Salad”, “Vitelotte”, “Red Emmalie”, “Blue Congo”, and three with bright flesh: “Bella Rosa”, “Lord”, “Tajfun”. The edible potato tubers were obtained from the plantation of the Carpathian State College in Krosno (latitude 21°46’E; longitude 49°42’N). The study was conducted during the vegetation period from 2020 to 2021.

Point 2. Line 23 content of: phenolic????

Response 2: Thank you for the remark, w prowadzonych badaniach wśród związków fenolowych zidentyfikowano cztery kwasy fenolowe (gallic, chlorogenic, caffeic, p-coumaric).

the sentence in the summary was supplemented with: ,,phenolic acids’’ (gallic, chlorogenic, caffeic, p-coumaric)’’

Point 3. Line 23-24 “The study focused on the content of: phenolic acids, total and reduced ascorbate, flavonols, 23 anthocyanins, and the antioxidant activity tested by means of ABTS, DPPH, and FRAP assays” does flow well. Also, no abbreviation in abstract.

Response 3: Thank you for the remark, questionable text content removed:

The wronf  text line 23-24;

,,The study focused on the content of: phenolic acids, total and reduced ascorbate, flavonols, anthocyanins, and the antioxidant activity tested by means of ABTS, DPPH, and FRAP assays’’.

The wrong text has been corrected line 23-24:

,,In all potato samples under study, 4 phenolic acids were identified: chlorogenic acid, gallic acid, p-coumaric acid, caffeic acid.

Point 4. Line 25 “highest levels of phenolic acids”…..please include range of this

Response 4: Thank you for the remark.The wrong text has been corrected line:

,,Total content of the individual identified phenolic acids was diversified and depended on the genotype of the cultivar and the tuber flesh colour, with coloured-fleshed potatoes having higher phenolic acid content as compared to bright-fleshed potato cultivars. The average concentration of phenolic acids in the samples was 89.19 mg∙100 g-1 FM, and for the individual phenolic acids identified were as follows: chlorogenic acid (86.19 mg∙100 g-1 FM), gallic acid (1.18 mg∙100 g-1 FM), p-coumaric acid (0.64 mg∙100 g-1 FM), caffeic acid (1.18 mg∙100 g-1 FM)’’.

Point 5. Line 26 please consider deleting “under study”

Response 5: Thank you for the remark , line , deleted the words  “under study”

The wrong text has been corrected line:

,,The predominant acid was chlorogenic acid, whose levels ranged 62.95 mg.100 g-1 FM to 126.77 mg.100 g-1 FM’’.

Point 6. Line 28 “was reported” ….please remove this

Response 6: Thank you for the remark , deleted the words  “was reported”

The wrong text has been corrected line:

,,Total content of the individual identified phenolic acids was diversified and depended on the genotype of the cultivar and the tuber flesh colour, with coloured-fleshed potatoes having higher phenolic acid content as compared to bright-fleshed potato cultivars. The average concentration of phenolic acids in the samples was 89.19 mg∙100 g-1 FM, and for the individual phenolic acids identified were as follows: chlorogenic acid (86.19 mg∙100 g-1 FM), gallic acid (1.18 mg∙100 g-1 FM), p-coumaric acid (0.64 mg∙100 g-1 FM), caffeic acid (1.18 mg∙100 g-1 FM)’’.

Point 7. Line 29 please remove this “According to a quantitative analysis”

Response 7:  Thank you for the remark , line, deleted the words  “According to a quantitative analysis”

Point 8.Line 34 please consider adapting “Pelargonidin-3,5-di-O-glucoside, peonidin-3,5-di-O-34 glucoside, petunidin-3,5-di-O-glucoside were the anthocyanins detected.”

Line 40-41 potentially improve health benefits????

Response 8:

corrected text:

,, In addition, 3 groups of anthocyanins were identified: pelargonidin-3,5-di-O-glucoside, peonidin-3,5-di-O-glucoside, petunidin-3,5-di-O-glucoside. Anthocyanins were not found in 'Lord' and 'Tajfun' varieties characterized by white tuber flesh. The dominant pigment was petunidin-3,5-di-O-glucoside, the average content of which was (25.95 mg∙100 g-1 FM), while the highest value was observed in the 'Vitelotte' variety (51.27 mg∙100 g-1 FM)’’.

Point 9. Line 59-61 please correct the grammar

Response 9: Thank you for the remark . The wrong text has been corrected line:

grammatically corrected text line

,,According to those authors, polyphenolic compounds in the diet can help maintain good body condition, prevent many diseases, and facilitate the treatment of existing disorders. According to Zhang et al. [2022] and Koszowska et al. [2013], a diet rich in antioxidants is beneficial to health as it decreases the incidence of cardiovascular diseases, diabetes, cancers, and osteoporosis. Research by Brown et al. [11] and Lachman et al. [6] showed that coloured-fleshed and -skinned potato cultivars also contain flavonol aglycones, the main group of plant phenols that are potent antioxidants’’.

Point 10.Line 70 how is “inhibition of RNA and linolenic acid oxidation” related to antioxidant activity? Please revise

Response 10. Thank you for the remark.

Incorrect entry:

,,Furthermore, the in vivo test conducted by Mulero et al. [13] showed high antioxidant activity of purple potato flakes, which involved the inhibition of RNA and linolenic acid oxidation. They also showed that the antioxidant properties of purple potato flakes enhance hepatic Mn-SOD, Cu/Zn-SOD and GSH-Px mRNA expression.’’

The wrong text has been corrected line :

,,In addition, a study by Han et al [19] on rats fed purple potato flakes showed that the flakes have antioxidant functions with respect to radical scavenging and inhibition of linoleic acid oxidation, and improved antioxidant potential in rats by increasing hepatic mRNA expression of Mn-SOD, Cu/Zn-SOD and GHS-Px.’’

Point 11. Line 75 “and smell substances”????

Response 11: Thank you for the remark.

Incorrect text has been removed :

,,Phenolic acids and their derivatives in potatoes are interesting as ingredients that enhance the precursors of flavour and smell substances, and as biologically active compounds that increase food quality.’’

Point12. Line 74-76 please revise

Response 12:  Thank you for the remark . Line text corrected:

,,Coloured-fleshed potatoes are recommended for consumption and production of fried and dried products, in particular due to good organoleptic characteristics (flavour, smell, texture, and colour), but also higher content of biologically active ingredients’’

Point 13. Line 77 “potato products offer” what do mean?

Response 13: Thank you for the remark.

erroneous words “potato products offer” corrected to  ,,potato product diversity’’

Point 14. 76-88 please improve.

Response 14: Thank you for the remark. Line incorrect text has been removed :

,,The introduction of coloured-fleshed potatoes into food processing can improve the potato products offer, which will stand out with not only their colour but also higher content of biologically active compounds. Therefore, the consumption of coloured-fleshed potato cultivars can potentially have more health benefits than traditional white-fleshed cultivars. It can have health benefits related to antioxidants, such as anti-cancer, anti-ageing, and anti-inflammatory properties. However, the source literature has rarely discussed the antioxidant potential of coloured-fleshed potatoes as compared to white-fleshed cultivars; therefore, the aim of this paper was to indicate the content of phenolic acids, anthocyanins, and antioxidant properties of coloured potato tubers, as well as to compare the content of these compounds to bright-fleshed potato tubers.’’

text corrected to:

,,Coloured-fleshed potatoes are recommended for consumption and production of fried and dried products, in particular due to good organoleptic characteristics (flavour, smell, texture, and colour), but also higher content of biologically active ingredients [4, 12]. The introduction of coloured fleshed potatoes into food processing can improve the potato product diversity, which will stand out with not only their colour but also higher content of biologically active compounds. Therefore, the consumption of coloured-fleshed potato cultivars can potentially have more health benefits than traditional white-fleshed cultivars. It can have health benefits related to antioxidants, such as anti-cancer, anti-ageing, and anti-inflammatory properties. Source literature reports that coloured-fleshed potatoes are beneficial to health. However, there has been little research on the influence of potato cultivars on the phenolic acid content and antioxidant properties of potato cultivars with coloured flesh and bright flesh. Therefore, the aim of this paper was to gain such knowledge and raise consumer awareness of the health benefits of coloured potato cultivars.’’

Point 15. Line 95 please include the soil and climatic characteristics of the area where the potatoes were cultivated. Also were they treated after harvesting before storage prior to the study? These are vital information to include.

Response 15: Thank you for the remark. A reviewer's comment was added to the text. Soil and climatic characteristics of the area where the potatoes were cultivated were given, information about the treatment was given.

Completed text

,,The raw material used in this study included tubers of 8 edible potato cultivars. Five cultivars were potato tubers with coloured flesh: “Rote Emma”, “Blue Salad”, “Vitelotte”, “Red Emmalie”, “Blue Congo”, and 3 with bright flesh: “Bella Rosa”, “Lord”, “Tajfun”. The description of cultivar characteristics is included in Table 1. The edible potato tubers were obtained from the plantation of the Carpathian State College in Krosno (latitude 21°46’E; longitude 49°42’N), Poland. The field experiment was conducted in 2021 in slightly acidic brown earth (pH/KCL 5.67) (WRB, 2014). The concentration of assimilable phosphorus was high (12.3 mg.100 g-1), potassium – medium (20.2 mg.100 g-1), magnesium – very high (19.5 mg.100 g-1), whereas the content of copper, manganese, iron and zinc in the soil was medium (Cu – 5.64 mg.100 g-1, Fe – 1,574 mg.100 g-1, Mn – 176 mg.100 g-1, Zn – 14.3 mg.100 g-1). The average content of humus in the topsoil amounted to 2.72%. The results of soil analysis were evaluated based on limit values established by the Institute of Soil Science and Plant Cultivation – National Research Institute in Puławy [Nawrocki 1990]. The field experiment was based on a randomised block design in three replicates. In the autumn, farmyard manure was applied at a rate of 25 t·ha-1 in addition to mineral fertilisers used at the following rates: 35 kg·ha-1 P (in the form of 46% triple superphosphate), 100 kg·ha-1 K (in the form of 60% potassium salt) and 80 kg N per 1 ha (in the form of 34% ammonium saltpeter), nitrogen applied in the spring. Potatoes were planted at 67.5 x 37 cm spacing in mid-April and harvested in September. The size of crop plots was 22.5 m2. To control weeds, a mixture of the herbicides Command 480 SC 0.2 dm3·ha-1 + Afalon Dyspersyjny 450 SC 1.0 dm3·ha-1 was applied 5–7 days prior to potato plant emergence. Potato blight was controlled using Ridomil Gold MZ 68 WG and Dithane 455 SC, and Colorado potato beetle was controlled by means of Apacz 50 WG and Actara 25 WG. Precipitation and air temperature in that period differed between the potato vegetation months. The year 2021 was warm with excessive precipitation in April, May and July, but significant deficits in July, August and September’’.

Point 16. Line 104 delete “For the study” and join the sentence with the preceding one. Also “slices (approx. 10 mm)” with what?  

Response 16: Thank you for the remark. The text has been supplemented with the reviewer's comments line 104. Removed "For testing" and merged the sentence with the previous one. Corrected the notation "slices (about 10 mm)" with what? 

The completed text reads.

        ,,Twenty tubers were collected at random, washed and dried, cut into 10 mm thick slices, then frozen at -35oC. The samples were dehydrated with the Sublimator (ZIRBUS Technology GmbH, Germany), then ground in a knife mill (Grindomix GM 200, Germany), and stored in a refrigerator in sealed plastic bags’’.

Point 17. Line 111 delete “means of”

Response 17: Thank you for the remark. Line 111, the reviewer's comment has been taken into account removed the word ,,means of’’

,,The concentration of each phenolic compound was measured by means of High-Performance Liquid Chromatography (HPLC),……’’

The correct notation reads:

,,For the purpose of identifying phenolic compounds in potato extracts, the High-Performance Liquid Chromatography (HPLC) …….

Point 18. Line 112 should be “as previously described [9].”

Response 18: Thank you for the remark. Line the reviewer's comment was taken into account and the erroneous text was corrected:

,,The concentration of each phenolic compound was measured by means of High-Performance Liquid Chromatography (HPLC), previously described in detail by Kazimierczak et al. [9]’’.

The correct notation reads:

,,For the purpose of identifying phenolic compounds in potato extracts, the High-Performance Liquid Chromatography (HPLC) was applied, as previously described [9]’’.

Point 19.  Line 113 -122 poorly described please improve

Response 19: Thank you for the remark.

incorrect notation

,,The concentration of each phenolic compound was measured by means of High-Performance Liquid Chromatography (HPLC),……’’

The correct notation reads:

,,For the purpose of identifying phenolic compounds in potato extracts, the High-Performance Liquid Chromatography (HPLC) …….

Point 20: Line 113-226 needs further improvements to enhance the descriptions

Response 20: Thank you for the remark. All of the reviewer's comments on improving the description have been addressed. The corrected text is:

Polyphenol Extraction and Identification

For the purpose of identifying phenolic compounds in potato extracts, the High-Performance Liquid Chromatography (HPLC) was applied, as previously described [9]. The sample preparation procedure involved the extraction of phenolic compounds from dehydrated potato samples (100 mg) with 80% methanol in plastic tubes by means of ultrasonic bath (10 min, 30oC). Then, the samples were centrifuged (10 min, 4000× g, at 4oC). Then, 900 µL of supernatant was injected into the HPLC vial, and 100 µL was injected on the HPLC column (Phenomenex, Fusion-80A, C-18, practical shape 4 µm, 4.6 x 250 mm, Nexera-i Plus Method Transfer, Shimadzu, Japan). Chromatographic separation was performed at 33±1oC with the following solvents: a mixture of water and acetonitrile (10% in phase A and 55% in phase B) at the flow rate of 1 mL min-1 used as a gradient solvent (1.00–22.99 min phase A 95%, 23.00–27.99 min phase A 50%, 28.00–28.99 min phase A 80%, 29.00–35.99 min phase A 80%, 36.00–38.00 min phase A 95%). The wavelength used for detection ranged from 270–360 nm. External polyphenol reference materials were used in the analysis, with the purity of 95.00–99.99%, from the Sigma-Aldrich company. Phenolic compounds were identified based on reference materials and retention times. The following wavelengths were used: 250 nm for phenolic acids (gallic, chlorogenic, caffeic, p-coumaric), 370 nm for flavonoids (quercetin-3-O-rutinoside, quercetin-3-O-glucoside, quercetin). The results were expressed in mg 100 g-1 FM.

Anthocyanin Extraction and Identification

Whole potatoes were washed in cold water, dried and cut into small pieces by means of a vegetable chopper. The study material of 800 grams was blanched for 3 minutes in 800 mL of water. The same amount of a mixture of water and hydrochloric acid (19/1, v:v) was cooled at 0°C for 3 hours and stored at 0°C, and stored at room temperature for at least 8 hours. In order to remove any solid material, the suspension was filtered through glass wool. The raw extract was applied on the Amberlite XAD-7 column. The column was carefully washed with water, and anthocyanins were eluated with a methanol/acetic acid mixture (19/1, v:v). The eluate was concentrated on a vacuum evaporator, then diluted in water and dehydrated. Anthocyanins were measured by means of the HPLC method according to the protocol previously described by Rodriguez-Saona et al. [18] subject to minor modifications. To 50 mL centrifuge tubes, approximately 50 mg of fine ground dehydrated potato tissue was added. Then, 10M HCl (hydrochloric acid; 2.5 mL) and pure 100% methanol (5 mL) were added, vortexed and soniced for 30 sec and left in dimmed light in a cool place for 15 min, stirred occasionally. Then, the sample was centrifuged at 6,000× g for 15 min. Then, 1 mL of the extract was transferred into the HPLC vial (Nexera-i Plus Method Transfer, Shimadzu, Japan) and 100 µL was injected onto the analytical column (Phenomenex, Fusion-80A, C-18, practical shape 4 µm, 4,6 x 250 mm). Two solvent systems were used. Solvent system A included water/acetonitrile/formic acid 87/3/10 (v:v:v), whereas solvent system B included water/acetonitrile/formic acid 40/50/10 (v:v:v). The gradient was as follows: 0 min 6% B, 20 min 20% B, 35 min 40% B, 40 min 60% B, 45 min 90% B and 55 min 6% B. The analysis was carried out with a wavelength of 530 nm, and the analysis duration was 18 min. Anthocyanins (pelargonidin-3,5-di-O-glucoside, peonidin-3,5-di-O-glucoside, petunidin-3,5-di-O-glucoside) were identified based on their pure reference materials and retention time.

Extract Preparation

The extraction of total phenols and anthocyanins was conducted based on the method by Yang et al. [19] subject to minor modifications. Two grams (g) of dehydrated potato sample was added to 15 mL of extraction solvent (95% alcohol) acidified with 1.5 N HCl (85:15, v/v). The tube was flushed twice. The obtained mixture was topped up to 50 mL and extracted for 12 hours at 4oC. The supernatant was collected into a 50 mL conical tube, and then centrifuged (4,000 rpm, 10 min, 4oC) in a centrifuge (ST16R, Thermo Scientific, Waltham, Massachusetts, USA). Polar lipids and other interfering compounds were removed by means of a method by Song et al. [7]. Eighteen mL of hexane was added to 6 mL of the obtained extract and mixed thoroughly prior to removing the hexane layer. This step was repeated six times until the hexane layer was removed completely. The chlorophyll-free extract was filtered with a 0.22 µM organic membrane and then used in the analysis of phenolic and anthocyanin content and the antioxidant activity.

Measuring Total Phenolic and Anthocyanin Content

Total phenolic content (TPC) was measured with Folin–Ciocalteu’s method with slight modifications. Two mL of the Folin–Ciocalteu reagent was added to 1 mL of the extract and mixed thoroughly. After 5 min, 2 mL of sodium carbonate was added (10 g/100 mL). The obtained mixture was mixed thoroughly and stored in darkness for 1 h at room temperature, and then absorbance was measured with a wavelength of 760 nm by means of the UV/Visible Spectrophotometer (DU800, Beckman Coulter). TPC was calculated using a calibration curve of gallic acid within the range from 1.045 to 10.45 µg/mL, expressed as a milligram of gallic acid equivalent per 100 g of fresh mass (mg GAE/100 g FM). Total anthocyanin content (TAC) was measured by means of a method described by Fuleki & Francis [Fuleki and Francis 1968]. The extract samples were mixed at a 1:20 (v: v) ratio with buffers: potassium chloride and sodium acetate with the pH of 1.0 and 4.5 respectively in separate tubes. After 15 min, absorbance of each solution was measured at a wavelength of 535 nm. Total anthocyanin content was calculated according to the following formula:

TAC (mg/100 g) = RG535 x V x N ¸ 98.2 ¸ m x 100

where RG535, V, N, 98.2 and m refer to, respectively: the spectrophotometric reading at a wavelength of 535 nm, extract volume, dilution factor, extinction coefficient and sample mass.

DPPH Radical Scavenging Activity

The ability of the samples under study to scavenge the stable free radical 2,2-diphenyl-1-picrylhydrazyl (DPPH, Sigma Aldrich Co., St. Louis, MO, USA) was measured by means of method [20]. Ten mL of distilled water (HLP 20UV, HYDROLAB, Straszyn, Poland) and 1.8 mL of a solution of 0.1 mM of DPPH in methanol (Sigma Aldrich Co., USA) were added to 2 g of the study sample. The obtained mixture was left in darkness for 1 h at room temperature. Then, the absorbance of the mixture was measured spectrophotometrically (UV-2600i Spectrophotometer, Shimadzu, Japan) at 517 nm with methanol as blind feed. The calibration curve was delineated within the range from 0.1 to 100 μgmL-1 of Trolox solution (Sigma-Aldrich, Co., St. Louis, MO, USA) in ethanol. The results were expressed as [μmol Trolox * g-1 DW].

ABTS Radical Scavenging Activity

The ABTS radical scavenging activity was measured with a method by Re et al. [21] with minor modifications described by Liao et al. [11]. At first, 4 µL of extract and 36 µL of absolute ethanol were poured onto a 96-well microplate with a flat bottom, and then 200 µL of ABTS radical solution was added. After mixing thoroughly and leaving the mixture in darkness for 6 min, absorbance was measured (Multiskan SkyHigh, Thermo Scientific, USA) at a wavelength of 734 nm The results were expressed as µM of Trolox equivalent per a gram of dry weight (µM Trolox *g-1 DW).

Ferric Reducing Antioxidant Power (FRAP) Assay

The FRAP assay was conducted according to the method described by Du et al. [22] subject to minor modifications. First, 0.2 mL of PBS and 1.5 mL of 0.3% (w/v) potassium ferrocyanide were added to 1.0 mL of extract diluted 10-times and incubated at 50oC for 20 min. Then, 1.0 mL 10% (w/v) of trichloroacetic acid was added. Next, the obtained samples were centrifuged for 10 min at 4,000 rpm at a temperature of 4oC. Afterwards, 2.0 mL of the supernatant was collected, and 0.5 mL of 0,3% (w/v) iron trichloride and 3.0 mL of distilled water were added. Absorbance was measured with the UV-2600i Spectrophotometer (Shimadzu, Japan) at 700 nm. The results were expressed as μmol Trolox g-1 DW.

Measuring Total and Reduced Ascorbate Content

Ascorbate (as ascorbic acid, AsA) was extracted and analysed in fresh potato tubers based on the method developed by Bartoli et al. [23] subject to minor modifications. Approx. 5 g of the potato sample under study was mixed with 5 mL of 5% metaphosphoric acid, and then ground in a pre-cooled ball mill (Teflon bowl with two zirconium balls) for 2 min. The homogenates were decanted into a 50 mL plastic centrifuge tube, and then another 10 mL of 5% metaphosphoric acid was used to flush the Teflon bowl and zirconium balls, and decanting was carried out into the same 50 mL plastic centrifuge tube. The homogenates were centrifuged at 10,000 rpm for 15 min at 4oC. The supernatant was poured into a cooled clean 50 mL plastic centrifuge tube and placed in ice. In order to measure reduced ascorbate content, 1,000 µL of 150 mM phosphate buffer (pH 7.4; 5 mM EDTA) and 200 µL of Milli-Q water (Millipore, Bedford, MA) were added to 200 µL of the supernatant. In the case of total ascorbate content, the same procedure was applied, the only difference being that 200 µL of Milli-Q water was replaced with 200 µL of 5 mM dithiothreitol. The reference materials were prepared in the same way as in the case of measuring reduced ascorbate content, the only difference being that 200 µL of each reference material was replaced with the relevant sample amount. The samples and reference materials were incubated at room temperature for 15 min in darkness. To each sample and reference material, o-phosphoric acid (100 µL) was added in order to neutralise dithiothreitol and acidify the solution for the analysis based on High-Performance Liquid Chromatography (HPLC). Then, the samples were filtered through 0.2 µm polyvinylidene fluoride filters (Chromatographic Specialties Inc., Canada) mounted on a glass syringe, which was flushed three times with methanol in-between the samples. Reduced and total ascorbate content was measured quantitatively by means of HPLC (Nexera-i Plus Method Transfer, Shimadzu, Japan) with the C18 analytical column (Luna 5 µL 150 × 4.6 mm i.d., Phenomenex, Torrance, CA). Oxidized AsA is the difference between total ascorbate content and reduced ascorbate content.

Point 21. Line 234 research objects???

Response 21: Thank you for the remark. The reviewer's remark about the correction of the word ,,research objects" has been included to ,,between variables''.

The text in this subsection ,,Statistical Analysis’’was changed to:

,,The results are presented as the mean of three independent measurements. All analyses were performed in Statistica v. 13.3 (StatSoft, Krakow, Poland). The significance of the means was examined post-hoc in Tukey’s test, which is part of the one-way analysis of variance (ANOVA). Differences of p <0.05 were considered significant. In order to describe the mutual relations between the variables under study, variable reduction techniques based on scaled heat maps made in the procedures of the R studio program were used.’’

Point 22. Table 2 where are the SD OR SEM not reported. Also what is the essence of those superscript numbers for? Doesn’t make the table nice and self explanatory. Please improve all tables in the manuscript

Response 22: Thank you for the remark .All the reviewer's comments were taken into account and the entries in all tables were corrected. The corrected tables have been included in the revised manuscript

Point 23. Results and discussions need improvements. Difficult to read the current form.

Response 23: Thank you for the remark. The description of the results and the discussion held has been corrected. The entire manuscript has undergone linguistic and grammatical correction. Evidence of improvement and inclusion of all comments is marked in yellow in the text.

We would like to thank the reviewer for his kind comments and ask him to accept the corrections in the text.

Round 2

Reviewer 3 Report

The authors made substantial changes in the manuscript according to the comment of reviewer.  However, the language and grammar is to be improved.

Author Response

Response to Reviewer 3 Comments

Manuscript ID: foods-1997809

Title: Phenolic acid content and antioxidant properties of edible potato (Solanum tuberosum L.) with various tuber flesh colours

Dear Reviewer 3,

We are very grateful for the thorough review of our manuscript. The reviewers' comments and suggestions, which have been implemented in the text, will significantly enhance its scientific value. The text has been revised in accordance with the reviewers' comments. Thank you for accepting our corrections and additions. We hope that the manuscript in its current form meets the requirements of the journal Foods.

Point 1. English language and style are fine/minor spell check required

Response 1: Thank you for the remark. The entire manuscript has undergone linguistic and grammatical correction. Evidence of the improvement can be found in the text under ,,Track Changes’’.

Reviewer 4 Report

The authors have improved the manuscript. 

Author Response

Response to Reviewer 4 Comments

Manuscript ID: foods-1997809

Title: Phenolic acid content and antioxidant properties of edible potato (Solanum tuberosum L.) with various tuber flesh colours

Dear Reviewer 4,

We are very grateful for the thorough review of our manuscript. The reviewers' comments and suggestions, which have been implemented in the text, will significantly enhance its scientific value. The text has been revised in accordance with the reviewers' comments. Thank you for accepting our corrections and additions. We hope that the manuscript in its current form meets the requirements of the journal Foods.

Point 1. English language and style are fine/minor spell check required

Response 1: Thank you for the remark. The entire manuscript has undergone linguistic and grammatical correction. Evidence of the improvement can be found in the text under ,,Track Changes’’.
